# Securing the Smart City Airspace: Drone Cyber Attack Detection through Machine Learning

**Zubair Baig** [1,*], **Naeem Syed** [1] **and Nazeeruddin Mohammad** [2]

[1] School of Information Technology, Deakin University, Victoria 3216, Australia; naeem.syed@deakin.edu.au
[2] Cybersecurity Center, Prince Mohammad Bin Fahd University, Dhahran 34754, Saudi Arabia; nmohammad@pmu.edu.sa
* Correspondence: zubair.baig@deakin.edu.au

**Abstract:** Drones are increasingly adopted to serve a smart city through their ability to render quick and adaptive services. They are also known as unmanned aerial vehicles (UAVs) and are deployed to conduct area surveillance, monitor road networks for traffic, deliver goods and observe environmental phenomena. Cyber threats posed through compromised drones contribute to sabotage in a smart city's airspace, can prove to be catastrophic to its operations, and can also cause fatalities. In this contribution, we propose a machine learning-based approach for detecting hijacking, GPS signal jamming and denial of service (DoS) attacks that can be carried out against a drone. A detailed machine learning-based classification of drone datasets for the DJI Phantom 4 model, compromising both normal and malicious signatures, is conducted, and results obtained yield advisory to foster futuristic opportunities to safeguard a drone system against such cyber threats.

**Keywords:** drones; criminal activity; machine learning; cyber attacks

## 1. Introduction

A smart city provides convenient and better quality services in large scale and interconnected urban dwellings. It can be defined as a confluence of a multitude of information and communication technologies to render services such as traffic management, logistics and delivery of goods. This facilitates automated, intelligent and adaptive service delivery to its citizens. During the COVID-19 crisis, the smart city global market was estimated at USD 741.6 Billion in 2020, projected to reach USD 2.5 Trillion in 2026 [1]. This rapid growth in digitally-enabled services during the COVID-19 crisis can be attributed to ready adoption of technology to enable remote access to services by the masses.

A smart city can comprise a range of traditional services that can be automated and be driven through Artificial Intelligence (AI)-based decision making. For instance, a traffic light can be presented with real-time traffic data that flows in from various locales in a smart city, to enable intelligence and adaptive signal transition timings, which improve traffic flow and lessen the chances of a jam. Similarly, a traditional electricity grid can be converted into a smart electricity grid to facilitate real-time energy utility information for end users as well as the grid operators. The smart energy segment is forecast to reach a global market of the value of USD 652.9 Billion by 2026 [1].

Unmanned aerial vehicles (UAVs), commonly known as drones, are an emerging facilitator of several smart city services. UAVs are controlled through a ground controller unit. They typically provide services such as observations of weather phenomena, aerial photography, product delivery and surveillance. Other examples of UAVs include remotely operated and unmanned flights, such as the S-100 Camcopter, designed to carry defense service payloads to remote and hard to reach locales [2]. Rapid proliferation of drones in the commercial market is evident with an estimate of its market share to reach USD 58.4 Billion by 2026 [3].

The benefits to both civilian as well as defence applications through adoption of drones are aplenty. For instance, monitoring of road traffic in a city grid through static cameras is limited by the scope of data collection. On the contrary, a drone can fly across a city's aerospace and gather data from numerous locales to help the city administrators identify traffic congestion points and take necessary action to enable smooth traffic flow. Such live feeds of data can be transmitted by the drone to a central smart city cloud service center for further processing (can be routed via the drone's Internet-connected ground controller). Smart city curb-side sensors can also be integrated with a central smart city facility comprising the cloud, drones, IoT sensors as well as smart vehicles, to facilitate a fully integrated platform of heterogeneous objects. Such a facility can be harnessed to enable the identification of available parking spots (rendered to a smart vehicle in real-time), identify emergency zones, which can be avoided by vehicles and assess threats from weather phenomena (including heavy rainfalls and tornado warnings) [4].

The benefits of having drones as part of a smart city landscape are aplenty, as described above. However, there is no holistic and comprehensive framework in place to identify, prevent or even detect cyber threats that are posed through the introduction of drones in a smart city's airspace. The intrusion of drones into no-fly zones poses a threat to public safety, compromised of a secure premise including penetrating an airport's airspace and posing threats to aircraft and airport operations, and the dropping of illegal goods (including delivery of unlawful products to prisons). Research and development in this space have seen significant advancement in recent times. For instance, Dedrone has proposed an AI-enabled portable drone detection unit (tower) for detecting unauthorized drone intrusions into no-fly zones through deployment of such monitoring towers at specific locales [5].

Whilst ongoing research in the domain of cyber attacks that involve drones is expanding very quickly, there is still a need to identify drone-based cyber attacks, assess the types of threats posed on a smart city's airspace and the impact of a drone-based attack to a city's economy. In this research, we present a novel scheme for identifying malicious drone behavior through a deep analysis of routine (normal) drone operations when obtained from actual drone flight data, reverse engineering of patterns of normal drone flight behavior, synthesis of drone attack traffic, specifically for hijacking, GPS signal jamming and DoS attacks and the adoption of machine learning techniques to classify the synthesized drone data into normal or malicious.

The rest of the paper is organized as follows. Section 2 provides a background to the study, including a discussion on the various types of threat models for drones. In Section 3, we present the acquired dataset representing normal drone data and an exploratory analysis approach for generating attack vector data. In Section 4, we present the intrusion detection framework for detecting malicious drone behavior through adoption of machine learning. The simulation setup and results analysis are provided in Section 5. The paper is concluded in Section 6.

## 2. Background

A smart city comprises a number of integrated components that enable its functioning and rendering of automated services to its citizens. Data generated by smart city components is facilitated for transmission across the ICT network to the next hop devices of the communication network topology. Data collection on a large scale is referred to as data volume. Analytics of collected data is contingent on the level of urgency and the computational capabilities of the controllers, edge devices and the geographic proximity of these to the central cloud platform. For instance, IoT data obtained from curb-side sensors will have to be communicated to the next hop edge device, whereupon further data processing can take place in the cloud. Data volume at a large scale is constantly communicated from resource-constrained devices, including drones, to a central cloud, for processing and subsequent decision making. Artificial Intelligence (AI) plays a significant role in accurate decision making from collected data in a smart city platform. In Figure 1, we present a smart city architecture.

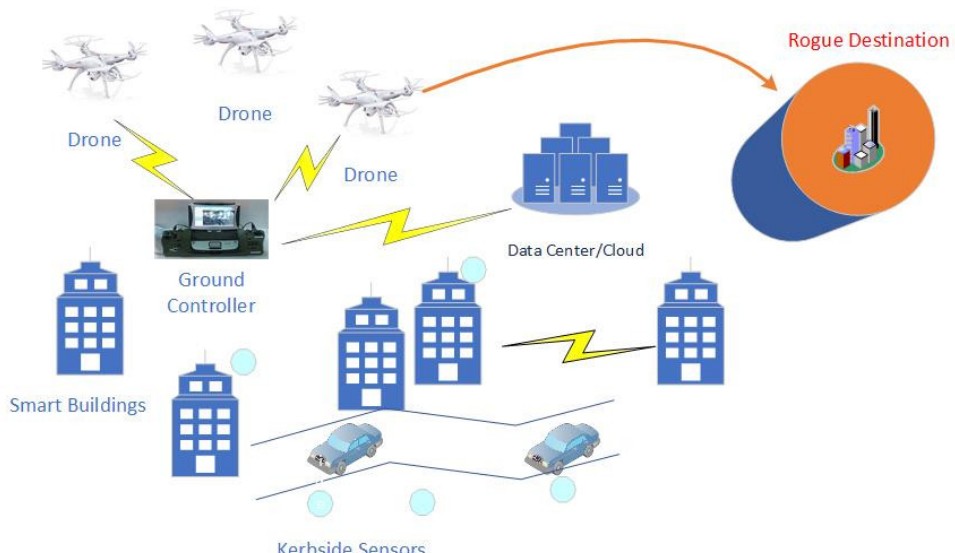

**Figure 1.** Smart city with drones—illustration of a potential threat.

*2.1. Drone Attack Models*

Drones, by design, do not comprise a fool-proof end-to-end security solution to reduce design and manufacturing costs. Common security gaps that persist within commercial drones are: vulnerability to firmware manipulation, lack of encryption of static data as well as communicated data (to the ground controller) [6].

Some drone manufacturers provide over-the-air (OTA) firmware updates analogous to mobile phone software patches (updates). Through such practice, vulnerabilities identified in drones post-purchase can be patched to avoid a compromise.

A software or firmware vulnerability in a drone can be exploited by the adversary through the modification of strings of data causing the drone to malfunction. Consequent adversarial actions can include flight trajectory changes and failure to encrypt flight logs [6].

On-device drone data is critical to operations, but also the transit from the drone to a ground-based controller in an unencrypted form could lead to a compromise. If the data are sensitive they could fall into adversarial hands and could subsequently be misused. Legacy network communication protocols would not encrypt drone data by default before it is communicated wirelessly to a ground controller. Additionally, the exposed vulnerabilities through unhardened firmware could lead to the disablement of encryption features, even if they exist on a drone.

In [7], the impact of drone-based threats to its operations have been categorized into the following:

1. Unavailability of a UAV;
2. Disruption of UAV operations;
3. Performance degradation and disconnection with ground controller;
4. Misleading GPS information;
5. Exposure of confidential UAV information;
6. Damage to infrastructure;
7. Compromised and misbehaving UAV.

The standard process to weigh in all parameters for a threat model, as proposed in [7] compromise: threat identification, severity estimator, likelihood of an attack, attack ranking and risk scores. The adversarial goals, as listed above, can be mapped to various threats that can be presented to a drone's software/firmware, a ground controller and the central cloud services that a drone connects to.

Additionally, an adversary may also pose a threat by hijacking a communication signal by either disrupting the signal or replacing the same with a malicious one. Consequently, the drone will operate outside its routine mode of operation.

Tran et al. [8] extend the specific operations risk assessment (SORA) [9] to include cybersecurity risks. They have proposed threat and harm extensions to the SORA methodology. Threat extension covers different cybersecurity threats that can occur to drones, which could make drone operations out of control. In other words, the threats are the main reason for drone-related hazards. SORA represents the potential outcomes of the Hazards as Harms. Cyber Harms primarily considers the privacy issues that can occur because of cyberattacks. It also considers physical and digital damages that could occur because of cyberattacks on drones. SORA includes two types of barriers: threat and harm barriers. Threat barriers prevent the Hazard incidence once a threat incident has occurred. Similarly, Harm barriers avert the Harm after a Hazard incident.

### 2.1.1. Denial of Service

In [10], a denial of service (DoS) attack was tested against the Parrot ANAFI Drone. The drone was connected to a Wi-Fi access point, that emulates a ground controller. The identified threat was the compromise of the Wi-Fi password, which can foster adversarial attempts to send falsified flight commands. With no knowledge of the password, the adversary may still be able to perform a deauthentication attack and attempt to crack the same. A third option could simply be an attempt to disconnect communication between the drone and the access point (ground controller).

A DoS attack can also be perpetrated through an adversary who is able to disrupt a drone-to-ground controller or ground controller-to-cloud communication channels by flooding the communication channel with a large volume of fictitious network traffic, consequently overwhelming the computational assets, and preventing them from continuing with routine operations. Such an attack can be carried out using a technique such as 'Low Orbit Ion Cannon' with TCP flooding attack enabled on port 80 if the communication between the ground controller and the cloud services adopts the TCP protocol. The Hping3 is command-line software that can also be deployed to carry out a TCP SYN flood attack (DoS) against a drone system asset [10].

A ground controller station can also be compromised by the adversary, which will then send a suspicious signal to mislead a drone or even cause a crash. Through an analysis of the receiver signal strength indicator (RSSI) values and by triangulating the numbers with neighboring drones and their data, the scheme proposed in [11] can prove to be resilient to DoS attacks.

### 2.1.2. Hijacking

Disruption of drone operations through modification of software can also be staged at the adversarial machine learning level. The drone quadrotor can be programmed to disrupt drone flight when it is subject to fictitious objects that can be observed through its sensors during flight, causing it to trigger a hazard avoidance routine and thus leading to a variation in flight trajectory. This can also be defined as a hijacking attack.

Quadrotors typically have a return to home (RTH) state, that may be induced by the adversary so as to cause abandonment of the next waypoint. Typically, a drone that runs low on battery or is disconnected from the ground controller would have the RTH state activated. However, the authors in [12] have identified four threat models, namely, jamming of the communication channel, obscuring the main sensor/mirror, duplicating the target image and disrupting the object tracker (hide or change object makeup). Subject to these threats, the drone can not simply be forced to change to an RTH state, but can also be misconfigured and forced to reach an incorrect waypoint. Deliberate attempt to present to the drone's vision sensors images, reflections or other attack vectors (visual) can be adopted by the adversary to cause such disruptions [12].

### 2.1.3. GPS Signal Jamming/Spoofing

GPS spoofing entails the presentation of inaccurate geo-location/coordinates to a UAV in-flight. GPS blueprints are widely available and can therefore easily be spoofed [13]. Such a malicious attack can be perpetrated simply by setting up a power amplifier and a transmission antenna that relays RF signals to a target. As drones are programmed to accept unencrypted GPS signals, spoofing of the data is a feasible attack and can lead to catastrophic results for the target drone. A varied coordinate signal could cause the drone to change its flight path and trajectory and may lead to a crash, with affected entities, including human citizens of the smart city, power lines, vehicles in commute and other ground objects [14].

Table 1 summarizes security property compromised, threat types and their impact analysis.

**Table 1.** Attack impact analysis.

| Level | Resources Affected | Motive | Impact |
|---|---|---|---|
| Hardware | Drone, ground controller | Sabotage, data exfiltration | Flight path, crash |
| Firmware | Drone, ground controller | Process and state tampering, Incapacitation | Crash, data loss |
| Software | Drone, ground controller, edge devices | Sabotage, compromise | Process disruption, Flight path/services crippled |
| Network | Communication channel | Data exfiltration, modification | Flight disruption, access denial |
| Process | Drone, ground controller, edge, cloud | Software malfunction | Flight disruption, data loss |

### 2.2. IDS Design

Drone-based attacker tactics against heterogeneous smart city ICT platforms are reliant upon the ability of the threat actor to intrude the drone firmware and/or the ground controller with the intent to either divert the same to an unwarranted locale, to cause it to crash or to modify its trajectory and make it observe and report phenomena back to a rogue command and control center (C2). Traditionally, intrusion detection for cyber physical systems can be categorized into signature-based and anomaly-based. For the former, the detection system has to be pre-configured with signatures of known attacks, which can subsequently be matched with live/observed drone data, possibly at the ground controller unit. Anomaly-based systems, on the other hand, are trained to identify normal/legitimate data flow, including network traffic which refers to routine drone behavior. Deviation from the norm is subsequently tagged as an attack. Contemporary intrusion detection system design comprises robust machine learning techniques that allow for correlation of multiple data streams that can emanate from the drone, ground controllers and edge nodes of a drone control system. Popular techniques that can be adopted for detecting adversarial attempts to penetrate the drone system include support vector machines (SVMs), deep learning, and extreme learning machines (ELMs) [15]. To place an intrusion detection system as an overlay on a ground controller would entail significant computing and storage usage.

The concept of training an intrusion detection system with known or malicious signatures, for subsequent detection, can be adopted without modification albeit the data itself will require massaging and preparation for presentation to the intrusion detection system during the training as well as the testing phases.

Whilst an intrusion detection system can be placed in a high performance device, including a ground controller to detect adversarial attempts to compromise a drone, the ability of such a system to function on resource-constrained devices, such as drones, is a challenge. We present an approach based on random forest machine learning, to enable a drone to re-train on classification data, with minimal overhead (see Section 5).

### 2.3. Machine Learning for Intrusion Detection

According to the Capgemini report, 61% of organizations confirm that they will not be able to identify critical threats without Artificial Intelligence [16]. AI-driven cybersecurity controls can detect a malicious attack before it achieves its malicious goals, predict future attacks through intelligent forecasting based on analysis of empirical data and present mechanisms for automated response to threats through generation and rendering of software patches to digital assets [16].

Machine learning can be defined as a category of Artificial Intelligence, wherein the notion of mathematical modeling of data is adopted to train the machine learning classifier. The classifier subsequently is subject to test data, which it classifies based on its developed capability during the training phase. Broadly, machine learning classifiers can be categorized into the following four categories:

- Supervised learning—the data presented to the machine learning classifier is labeled as per its class definition. For instance, in a *drone attack* the label can be placed for those data samples (rows of data) that represent an attack vector. Similarly, routine drone flight data can be categorized as *normal*, which can serve as the second label for data samples. During the testing phase, data samples are presented to the trained classifier (model) without labels, and performance measurements of the classifier are measured through comparison of its classification outcomes to the actual class lables of the test dataset.
- Unsupervised learning—the data presented to the classifier are unlabeled, and the classification procedure in itself follows the process of clustering similar data samples into a given cluster and through differentiation at the inter-cluster levels.
- Reinforcement learning—the concept is based upon producing a 'rewarding function', which produces an optimal or a near-optimal classification of data samples, without the dependence upon labels or supervision. Typical reinforcement learning algorithms adopt Markov decision models to assess input data samples to attain the highest cumulative reward, when the classification is performed. This concept can be combined with supervised learning (for labeled data samples) to enhance the overall accuracy of the classifier.

Popular machine learning classifiers for drone applications include Naive Bayes, support vector machines (SVMs) and random classifiers. Random forest classifiers are ensemble-based classifiers known for their robustness in image classification. In [17], random forest classifiers are adopted for classification of images captured by a drone to identify vegetation in remote sensing fields. The first step of the classification is to adopt a bootstrap strategy wherein, nearly two-thirds of the training data samples are consumed to produce a decision tree. The remainder data samples are named out-of-bag data, which are subsequently used for inner cross validation of the trained random forest decision tree model, for accuracy.

Support vector machines (SVMs) are supervised machine learning algorithms that belong to the family of linear classifiers. The objective of the SVM training algorithm is to build a hyperplane in an $N$-dimension space that maximizes the margin between the two classes of data. Hyperplanes are typically decision boundaries that enable the distinguishing of data points of one class from another.

The Naive Bayes (NB) classifier is a simple probabilistic technique that is based on the concept of the Bayes theorem. It functions by assigning a posterior probability to a data sample for belonging to a class, $Y$, based on the a priori training of the classification algorithm on a dataset. Attributes are assumed to be independent of each other, i.e., feature dependence is not a criteria for training and testing. NB classifiers are known for their high accuracy in classifying string data.

Supervised learning techniques have been previously adopted to identify cyber threats in a drone system. In [18], a framework is presented for the classification of drone data into malicious or normal through adoption of standard drone data for presenting to a post-incident machine learning classifier, in order to infer a cyber criminal activity as part of digital forensic investigations.

## 3. Data Acquisition and Attack Vector Generation

Considering the wide spread adoption of drones for smart cities, to undertake operations and render timely services, associated cyber threats need to be highlighted. Once an intruder is successful in gaining access to a UAV, it can carry out other attacks such as data theft (of sensitive video and image recordings from drone), data corruption, Denial of Service (DoS) or signal jamming, with a motive to sabotage a UAV in flight or to modify its flight path and fly it to an unwarranted waypoint. A compromised drone can be forced to crash through GPS spoofing techniques or by deliberately powering-off the motors in an attempt to rapidly drop its altitude, with an intent to incapacitate (or completely damage) the drone.

Detecting anomalies in a drone through the analysis of its on-device data is, therefore, imperative to foster a secure drone flight and to prevent a cyber attack from occurring. To our knowledge, there does not exist a robust and labeled flight dataset for UAVs to introduce an anomaly detection exercise, and therefore, we proposed a methodology wherein abnormal (outlier) events are found through analysis of flight logs of a DJI Phantom drone are reverse engineered to construe attacks.

Several references discuss the presence of anomalies in drone data. We examine references [19–21] to interpret the possibility of anomalous flight patterns in drones. In [21], motor temperature monitoring is undertaken to identify beyond-normal patterns. In [19], drone in-flight faults that could cause a crash have been highlighted. To detect such faults, motor speeds and drone altitudes are examined to identify potentially anomalous events in drone logs. Similarly, drone log analysis techniques described in [22] illustrate the use of drone flight logs for detecting a drone crash. The proposed scheme adopts the analysis of data obtained from the accelerometer, drone motor and altitude, to detect a crash. Based on this, it can be concluded that flight logs are an important source of information, which comprise both normal and anomalous drone flight operation data.

As part of our proposed attack model, we first identify the potentially anomalous flight operations using exploratory data analysis and then identify potential scenarios that might closely match cyber attacks.

### 3.1. DJI Phantom 4 Data

The DJI Phantom 4 drone dataset was acquired from VTO Labs [23], comprising DAT files of flight logs, obtained from various flights undertaken by a single drone. The dataset includes 40 flight logs. However, only 18 logs could be adopted for our experiments due to the presence of inconsistencies and errors such as missing parameter values and lack of GPS data or the occurrence of truncated files, which were consequently discarded. These flight logs comprised information recorded by the main components of the UAV, including the flight controller, gyro stabilizer, on-board flight computer communication system, power supply, GPS modules and the likes [24]. The DAT files were processed and converted into CSV files using the CSVView and DatCon tools as also adopted in previous studies [25]. The sampling rate chosen in the DatCon tool to extract the CSV formatted file, was set at 10 Hz.

CSV files thus obtained comprised 289 fields (column labels). Categories of data available in these flight logs are listed as follows:

- State Signals;
- Time-Series Signals (including):

  – Air Speed,
  – ATTI_MINI (Attitude Mode),

- Battery Info,
- Batter Status,
- Clock,
- Compass Filter,
- Controller,
- GPS,
- IMU_ATTI (inertial measurement unit),
- Motor and Motor Control,
- osd Data (on screen display),
- RC_info (radio controller information).

State signals are categorical fields indicating the state of the drone whilst in flight. In contrast, the time-series signals are numerical fields with some observed value of various fields related to its flight. In this work, we only study the time-series fields to avoid unpredictable behavior of machine learning algorithms, given the variability of time series data (i.e., entropy). Specifically, the flight logs from the following file path within the DJI Phantom memory system were considered: "Drone_Forensics > DJI_phantom_4 > df005_DJI_Phantom_4 > 2018_June > flight_logs > flight_logs.zip". To perform exploratory analysis and classification tasks, only data files that contained the full set of 288 features were considered. Some of the important features related to flight logs are listed in Table 2.

**Table 2.** Important flight data features for intrusion detection tasks.

| Field | Description |
|---|---|
| Clock:offsetTime | Time since the start of the flight |
| 'IMU_ATTI(0):Longitude' | Drone's geographic east-west location on Earth's surface |
| 'IMU_ATTI(0):Latitude' | Drone's geographic north-south location on Earth's surface |
| 'IMU_ATTI(0):relativeHeight:C' | Altitude of the drone |
| 'IMU_ATTI(0):roll:C' | Left-right movement angle of the drone horizontally |
| 'IMU_ATTI(0):pitch:C' | Forward and backwards movement angle of the drone |
| 'IMU_ATTI(0):yaw:C' | Rotation around the drone's central axis |
| 'IMU_ATTI(0):distanceTravelled:C' | Distance travelled by the drone from the start point |
| 'Controller:ctrl_throttle:D' | Throttle given to the drone usually for providing lift. |
| IMU_ATTI(0):numSats | Number of satellites that the drone is connected to |
| Controller:motor_average_speed:D | Average speed of the drone's four motors. |
| osd_data:navHealth | The navigation health of drone in terms of number of GPS satellites it is connected to. Value of zero indicates unreliable GPS data and value of five indicates good signal reception. |
| RC_info:frame_lost | This field indicates the connectivity between radio controller and the drone. |

**Table 2.** *Cont.*

| Field | Description |
|---|---|
| ctrl_throttle:D | This indicates the throttle given to the drone from the user. |
| BatteryStatus:volLevel | This field indicates the voltage level of the battery. |
| Motor:Speed:RFront | Drone's right front motor speed |
| Motor:Speed:LFront | Drone's left front motor speed |
| Motor:Speed:LBack | Drone's left back motor speed |
| Motor:Speed:RBack | Drone's right back motor speed |

*3.2. Exploratory Analysis—Discerning Malicious Outliers*

In order to conduct an exploratory analysis, we first adopted the commonly used clustering algorithm, namely, unsupervised Gaussian mixture models (GMM) for cluster and outlier analysis as shown in Figure 2. This procedure was undertaken to identify potential anomalous points in the dataset. For the exploratory analysis, we only considered a subset of flight logs. Instead of using Euclidean distances, GMM uses probability distribution of various points (of the feature-value tuples of the dataset), to identify various data distributions that exist in the dataset. The advantage of using GMM is that it considers data points that belong to the same cluster and which follow a Gaussian distribution. Hence, each distribution is defined as a unique cluster. Figure 3 presents the various clusters formed in the partial dataset, which also contains outlier samples. Further analysis of the flight path data for this data subset yields the finding that there are anomalous flight events in flight log 19, as shown in Figure 3.

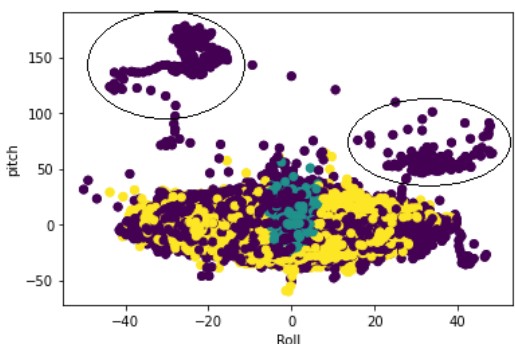

**Figure 2.** Pitch vs. roll in flight trajectory clusters obtained from GMM.

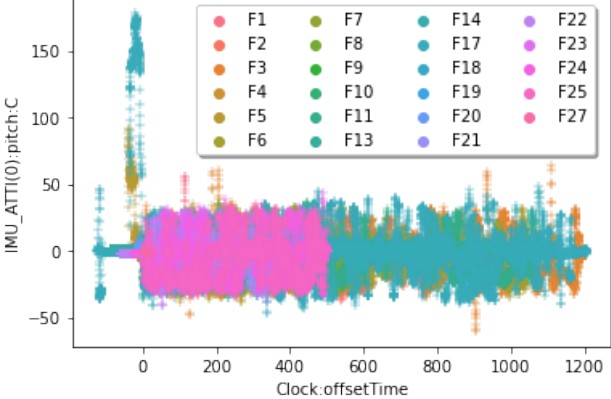

**Figure 3.** Identifying the flight number with abnormal pitch values.

Further analysis of flight19 logs using the CSVView tool indicates more anomalous events during the flight. The data revealed that an anomalous event occurred in the initial stages of the flight operations, which comprised a sharp increase in the pitch angle and a sharp descent of the drone along with a low/zero thrust to the drone, as shown in Figure 4. The sudden drop in the flight altitude can also be attributed to a cyber attack wherein an intruder deliberately hijacks the drone and suddenly impedes throttle to cause the drone to descend steeply and crash and get destroyed.

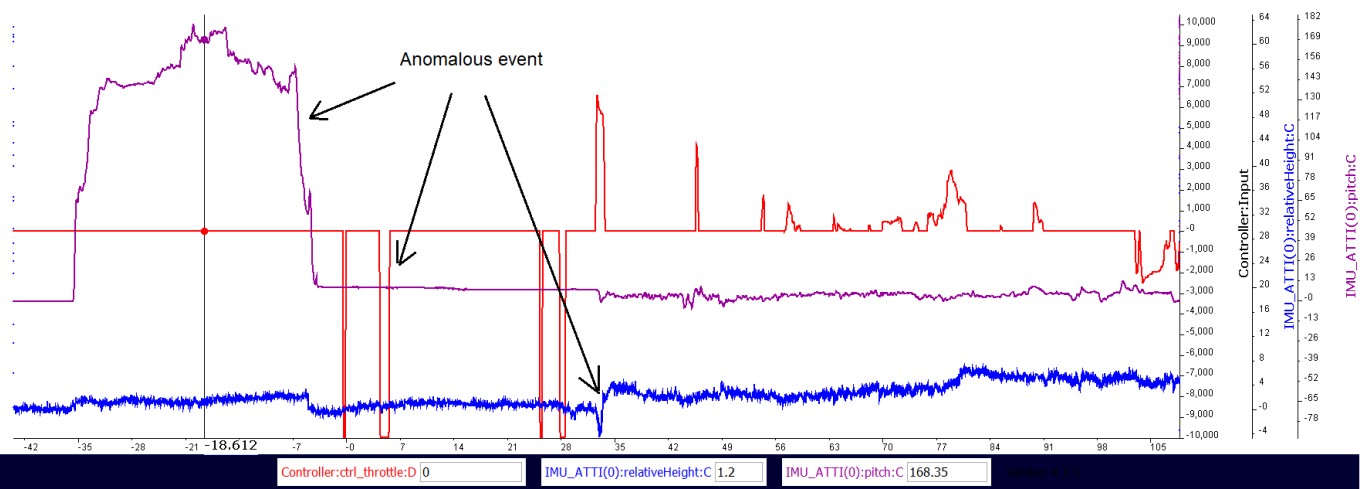

**Figure 4.** Abnormal operations in flight-log-19.

The GPS jamming attack comprises a disruption of GPS signals through jamming of signals (radio) that are transmitted and received between a drone and a ground controller unit. Figure 5 shows the time window of the flight operation wherein the GPS signals become unreliable. A GPS jamming attack results in a similar situation wherein, GPS signals can be disrupted to cause the drone to loose the ability to identify its flying coordinates, and possibly trigger a Return To Home (RTH) routine [14].

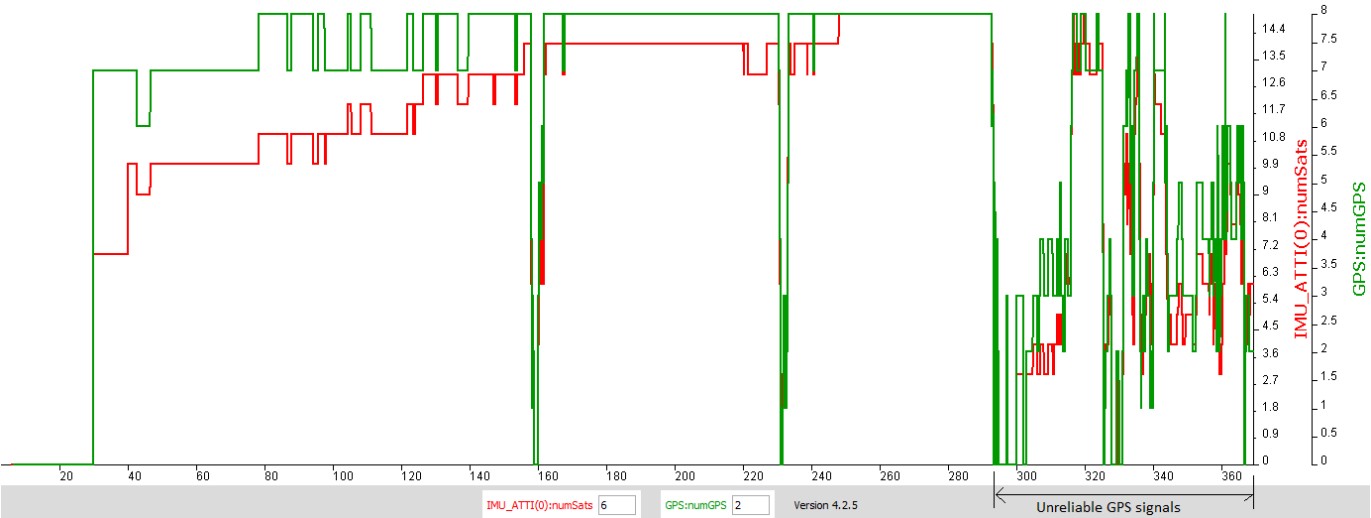

**Figure 5.** Unreliable GPS signal observed in the drone.

A DoS event is the occurrence of a sudden loss in the power supply to the drone motors/rotors (hijacking). The thrust to the drone is provided by its four motors (two in front and two in the back), in a typical quadcoptor model. Through switching of these motors, the drones are maneuvered and its height is controlled.

In regard to hijacking attacks, a sudden drop in motor speeds can occur when an intruder successfully gains access to the drone's ground controller or is successful in hijacking a remote asset such as an intermediary device that the ground controller is communicating with, and can thus deliberately power off the motors. This potential anomalous event is illustrated in Figure 6.

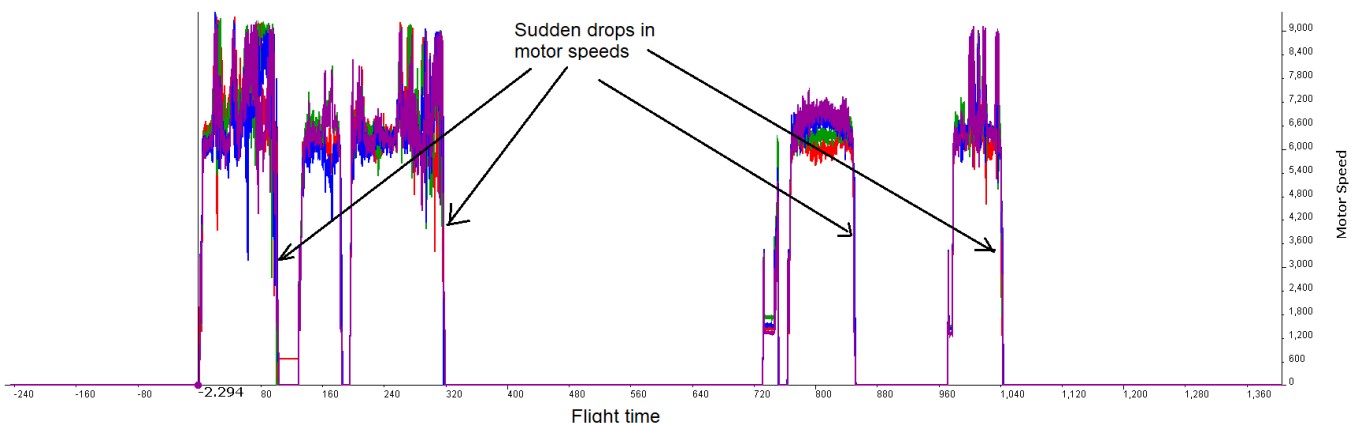

**Figure 6.** Sudden drop in motor speeds observed in the drone.

A variant denial of service (DoS) attack scenario causes the data communication flow between a ground-based radio controller and resulting in significant communication loss. In-flight parameters, namely, 'osd_data:connectedToRC' and 'RC_Info:frame_lost:D' capture the status of the connection between the radio controller and the drone. During in-flight drone operations, situations may arise wherein communication between a radio controller and the drone is lost under certain natural weather phenomena. We argue that in the presence of a DoS attack, a similar situation may arise due to the communication between the drone and controller being lost, not as a consequence of a natural phenomena but rather through a malicious event of sabotage perpetrated by the adversary. Hence, we identify the instances of the dataset wherein communication is disrupted, to mark a potential anomaly representative of a DoS attack. It may be noted that a DoS attack even for legacy systems is hard to distinguish from flash crowds comprising legitimate network traffic, though their impact on the victim device (drones in our case) is the same [26].

For routine operations, DJI drones communicate with the radio controller using some form of wireless communication technology (i.e., ocusync or other versions also use Wi-Fi). If the communication channel is under attack the drone can lose communication with the radio controller. The logs record the communication health with the radio controller using the rc-info-frameloss variable. We ignore arbitrary frame loss for the study, however, we label a continuous frame loss as being part of adversarial dirsuptions, that can refer to a DoS attack.

In addition, we also replicate the frame loss events obtained from flight logs and increase the number of such instances wherein a frame loss has occurred, to add a range of random anomalous (DoS) events to the log file (and to obtain a good balance between normal and anomalous data samples). Similarly, instances where the connection between the radio control (RC) and the drone is lost (indicated by frame_loss) are also marked as a DoS attack, as shown in Figure 7. Variant versions of DoS attacks (including stealthy attacks) were not considered for our experimental analysis.

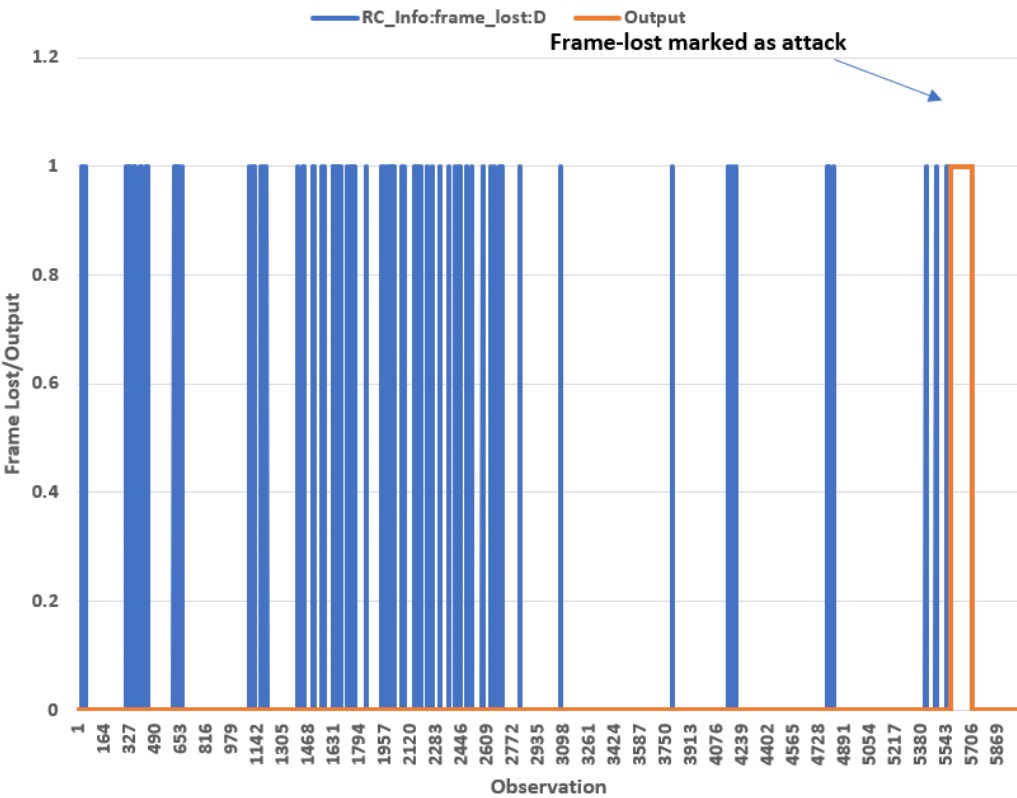

**Figure 7.** Continuous frame loss due to DoS attacks against the drone.

As shown in Figure 8, the parameter rc_connect changes the state to zero, indicating a DoS attack. In Figure 7, blue lines indicate frame-loss recorded in the flight logs and orange lines indicate the label given to these instances.

| osd_data:compassError | osd_data:connectedToRC | osd_data:lowVoltage |
|---|---|---|
| FALSE | Connected | low |
| FALSE | Connected | low |
| FALSE | Connected | low |
| FALSE | DisConnected | low |
| FALSE | DisConnected | low |
| FALSE | DisConnected | low |
| FALSE | DisConnected | low |
| FALSE | DisConnected | low |
| FALSE | DisConnected | low |
| FALSE | DisConnected | low |
| FALSE | DisConnected | low |
| FALSE | DisConnected | low |
| FALSE | DisConnected | low |
| FALSE | DisConnected | low |

**Figure 8.** Drone and radio control disconnected.

## 4. Proposed Intrusion Detection Framework

The proposed framework used in this work is presented in Figure 9. The first stage comprises obtaining the flight logs as DAT files from the VTO labs drone dataset repository. The DAT files were then converted into CSV files using the tools and processes mentioned in [27]. The extracted dataset was then analyzed for various anomalous data points that match the attack scenarios listed, namely, GPS signal loss, jamming and DoS attacks.

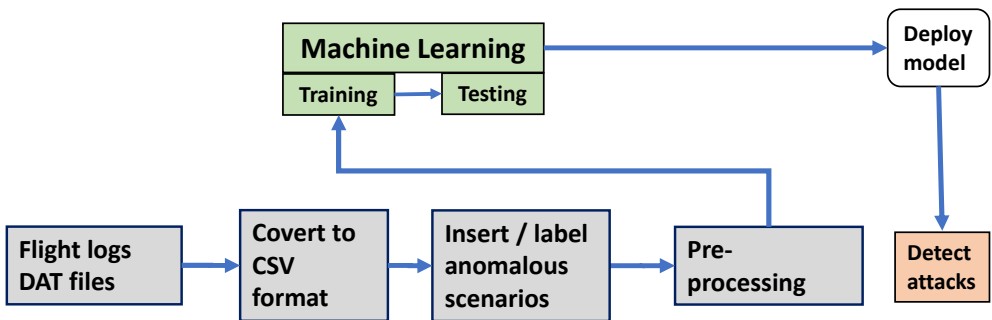

**Figure 9.** Framework to detect malicious attacks in UAVs.

Based on the types of cyber attacks that target UAVs, the abnormal data points were labeled as attack and remaining data points as normal. For the DoS attack scenario, additional data points that replicate frame loss were inserted into the dataset that reflect the drone state during the attack. The various flight logs were then combined to form a single dataset with multiple flight scenarios. From the dataset, features that highlight the flight path were extracted for classification tasks. The features extracted were: 'IMU_ATTI(0):Longitude', 'IMU_ATTI(0):Latitude', 'IMU_ATTI(0):relativeHeight:C', 'IMU_ATTI(0):roll:C', 'IMU_ATTI(0):pitch:C', 'IMU_ATTI(0):yaw:C', 'IMU_ATTI(0): distanceTravelled:C', 'RC_Info:frame_lost:D', 'Motor:Speed:RFront', 'Motor:Speed:LFront', 'Motor:Speed:LBack', 'Motor:Speed:RBack', 'osd_data:navHealth', 'IMU_ATTI(0):numSats' and 'Output'. The pre-processing stage then further removes data points with valid values, such *'NAN'* are removed from the dataset before the classification task.The dataset details are listed in Table 3.

**Table 3.** Dataset description.

| Categories | Dataset | Total Flights | Types of Attacks Included |
|---|---|---|---|
| Normal | 97255 | 18 | - Deliberate motor shutdown (all four motors speed = 0)<br>- DoS attacks (frame loss = 1)<br>- GPS jamming (navhealth = 0) |
| Attack | 14590 | | |

navHealth is a log feature that indicates the health of communication messages between a drone and GPS satellites. Anomalous navHealth values can be attributed to a drone's inability to communicate properly with satellites, and this can also potentially be attributed to a GPS jamming attack wherein, radio signals are impeded to prevent routine drone to satellite communication, and thus disrupting flight operations. Figure 5 presents the anomalies as obtained through analysis of the navHealth parameter values.

Once the pre-processing is completed, the dataset is then fed to a machine learning algorithms such as random forest, navie bayes, linear regression and SVM. The best performing model is then selected for deployment for intrusion detection in drones.

## 5. Simulations and Analysis

### 5.1. Setup

The exploratory analysis and machine learning tasks were performed on a machine with a Windows 10 Laptop, 16 GB RAM running on an 11th Generation Intel(R) Core(TM) i7-1185G7 3.00 GHz. The simulations and machine learning-based data clustering tasks were performed using Python Scikit-learn libraries [28].

### 5.2. Data Split and Metrics

The dataset for the DJI Phantom 4 was split into a 80–20 configuration with 80% used for training and the remainder 20% for testing of the trained models. The splitting method chosen was based on the Pareto principle wherein an 80/20 rule is adopted to split the dataset into training and testing samples. The following performance metrics were considered, and associated values were generated by the simulator:

- Accuracy: $\dfrac{TP + TN}{TP + FP + TN + FN}$ is the number of instances correctly classified given all the instance predictions, where true positive is $TP$, true negative is $TN$, false positive is $FP$ and false negative is $FN$.

- Recall: $\dfrac{TP}{TP + FN}$ indicates the number of $TP$s compared to the positive class instances. A low recall value represents a high number of attack instances missclassified as normal.

- Precision: $\dfrac{TP}{TP + FP}$ indicates the number of TPs predicted correctly. A low value of precision represents a high proportion of false positives whereby normal instances are marked as attack instances.

- Area under the curve (AUC):

$$\int_0^1 TPR(x)\, dx$$

where TPR is the true positive rate and $x$ is the false positive rate. Values range between 0 and 1.

Furthermore, the performance of the random forest classifier was evaluated for various values of the max-depth parameter and variable number of estimators. Max-depth and number of estimators are hyperparameters of a random forest classifier that can tuned for identifying the best performing settings. Max-depth controls the tree depth of the random forest, which, in turn, controls the complexity of the generated decision tree. Increasing the tree depth can increase the complexity of the decision tree and can also impact training time. A high value of max-depth can cause the random forest to over-fit it and, hence, it is not a recommended practice [29]. Through our studies, we explored the values of max-depth in the range of two to six and considered the number of estimators to control the number of trees generated for the random forest. The value explored was between 5 and 15.

### 5.3. Analysis

Results obtained through experiments yielded an outcome that showed superior classification accuracy of the random forest classifier in detecting the three UAV attacks presented in this work. Table 4 shows the impact of choosing different values for the max-depth parameter and the resulting best detection results were obtained with a setting, max depth = 6. In contrast, the results obtained for varying the number of estimators presented in Table 5 and Figure 10 show that the accuracy of the model first increases with the number of estimators and then reduces after the value of nine is reached. Hence, we can conclude that the optimal number of estimators for random forest in detecting UAV attacks is *nine*.

**Table 4.** Impact of max_depth on accuracy, precision and recall rates for the random forest classifier.

| max_depth | Accuracy | Precision | Recall | Training Time |
|-----------|----------|-----------|--------|---------------|
| 2 | 0.9517 | 0.8767 | 0.7518 | 0.2288 |
| 3 | 0.9796 | 0.9818 | 0.8666 | 0.2599 |
| 4 | 0.9782 | 0.9979 | 0.8421 | 0.3401 |
| 5 | 0.9914 | 0.9930 | 0.9436 | 0.3846 |
| 6 | 0.9947 | 0.9960 | 0.9653 | 0.4455 |

**Table 5.** Impact of number of estimators on accuracy, precision and recall rates for the random forest classifier.

| n_estimators | Accuracy | Precision | Recall | Training Time |
|--------------|----------|-----------|--------|---------------|
| 5 | 0.9636 | 0.9847 | 0.7448 | 0.1424 |
| 7 | 0.9651 | 0.9841 | 0.7567 | 0.1862 |
| 9 | 0.9784 | 0.9759 | 0.8631 | 0.2544 |
| 11 | 0.9615 | 0.8843 | 0.8264 | 0.3371 |
| 13 | 0.9763 | 0.9848 | 0.8393 | 0.3557 |
| 15 | 0.9757 | 0.9896 | 0.8309 | 0.4062 |

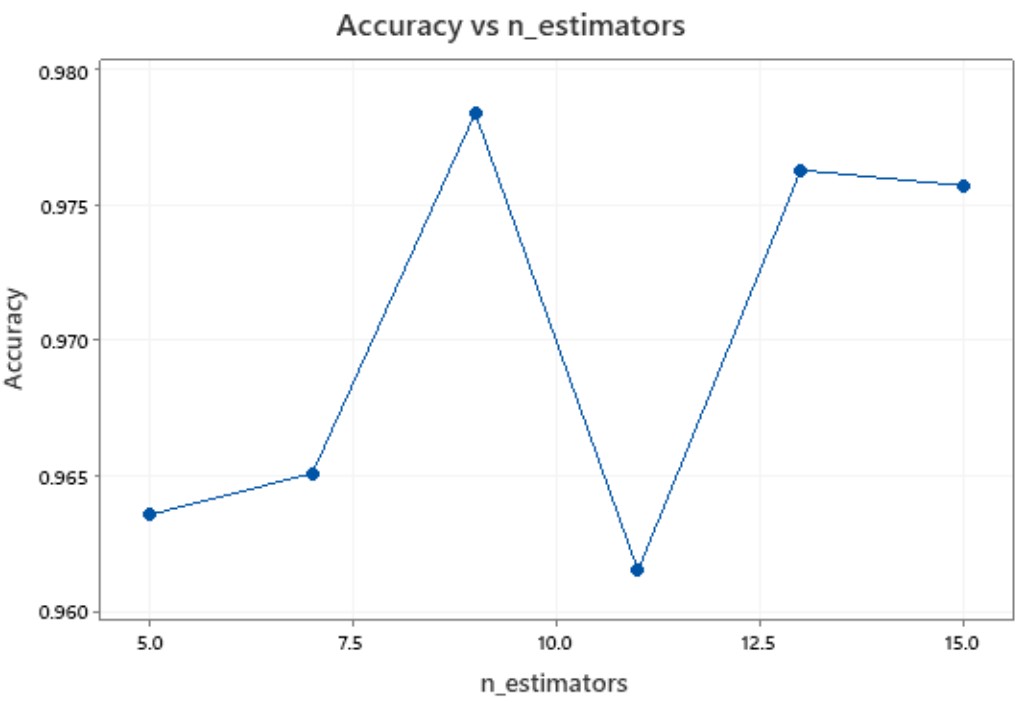

**Figure 10.** Impact on random forest accuracy for increasing numbers of estimators.

Similar observations can be made from the receiver operating characteristics (ROC) curve that illustrate the model performance, as shown in Figures 11 and 12. The ROC curves plot the false positive rate against the true positive rate indicating the discrimination capability between classes for the generated models. A higher area under the curve indicates better capability of models to distinguish between UAV attacks and normal UAV flight operations. Comparing the two hyper-parameters, namely, the max depth and the number of estimators, it is found that the max-depth increases the ability of the generated/trained model to differentiate between the attack and normal classes but may also

lead to overfitting. However, the number of estimators with a value of nine showed the best differentiation capability between normal and malicious drone flight logs.

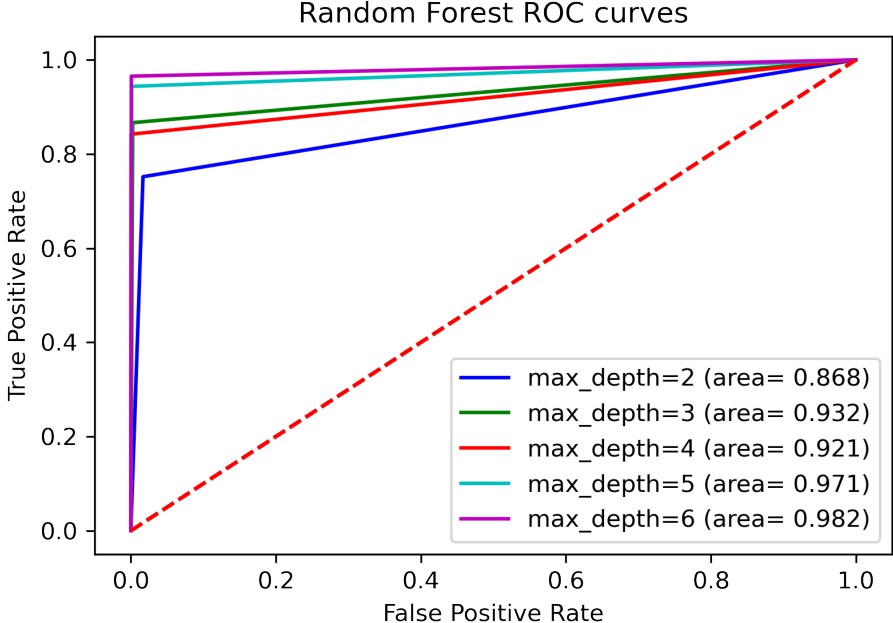

**Figure 11.** ROC curve comparing the performances with varying values of max_depth.

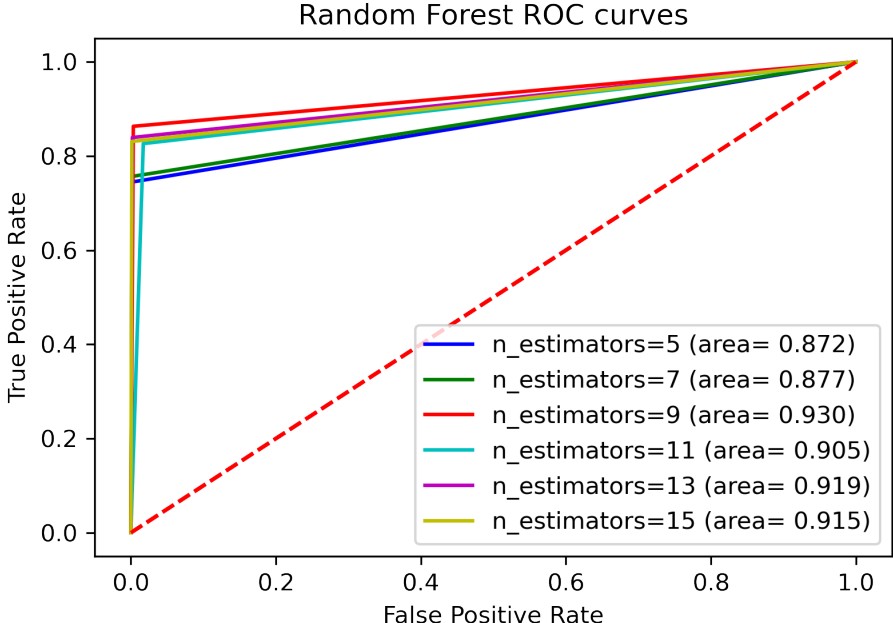

**Figure 12.** ROC curve comparing the performances with varying values of numbers of estimators.

The random forest classifier also outperformed other classifiers, namely, Naivee Bayes (NB), linear regression (LR) and support vector machines (SVM). Table 6 presents the comparison of individual classifier performances against the random forest scheme. The results show that random forest with a max-depth set to three and number of estimators set to nine obtained better accuracy, precision and reasonably good recall rates. Especially with large differences noted between the normal and attack categories, the precision and recall results are essential metrics to justify classifier performance in terms of the number of UAV attacks and normal instances to be correctly labeled as belonging to attack and normal classes,

respectively. The results also show that random forest had very high precision values when compared to other models indicating it was able to detect most attacks and normal instances correctly with low false positives. On the other hand, the other classifiers had higher recall rates when compared to random forest, indicating that certain attack instances of the UAV flight were classified as normal. The reason for this could be that the attacks, such as frame loss, occuring during a DoS attack may also occur due to various other non-adversarial causes, including the longer distance of the drone (in-flight) from the ground controller, terrain specifications and weather conditions. Hence, it is challenging to detect attack instances when drones are operated in difficult conditions, and to differentiate the same against naturally caused malfunctioning.

To further compare the generalization performance of classifiers, we obtained flight logs from a second DJI Phantom 4 drone obtained, from the VTO labs repository [23] comprising eight usable log files. After following a similar labeling process as was undertaken for the first dataset, we compared the performance of the four classifiers on the combined dataset, with results illustrated in Table 7. A comparison of classifier accuracies for both a single drone as well as a combined drone dataset is presented in Figure 13. For the combined datset, the results show that the accuracy of the RF classifier slightly drops and slightly increases for other classifiers. However, looking at the precision and recall values in Table 7, all classifiers yielded a degraded performance. This could be due to an increased number of normal data samples when compared to attack samples in the combined dataset. These results also indicate that the RF classifier still outperforms the other classifiers in terms of accuracy and precision rates, making it more suitable for detecting DoS and GPS jamming attacks.

Finally, looking at the time required to build the models by various classifiers, Naive Bayes yielded the lowest training time and SVM yielded the highest with just 5000 data samples. This is an important aspect to be considered when developing models for UAV intrusion detection. With low computation capabilities on a typical drone, on-device rapid training on smaller datasets is thus more realizable.

**Table 6.** Comparison of various machine learning models for detection rates.

| Classifier | Accuracy | Precision | Recall | Training Time |
|---|---|---|---|---|
| RF | 0.9784 | 0.9759 | 0.8631 | 0.2544 |
| NB | 0.8595 | 0.4930 | 0.9958 | 0.0306 |
| LR | 0.8595 | 0.4930 | 0.9958 | 0.4992 |
| SVM (5000 samples) | 0.848 | 0.4773 | 0.9856 | 130.086 |

**Table 7.** Comparison of various machine learning models for combined dataset from two drones.

| Classifier | Accuracy | Precision | Recall | Training Time |
|---|---|---|---|---|
| RF | 0.9686 | 0.8919 | 0.7905 | 1.3088 |
| NB | 0.8837 | 0.4679 | 0.9639 | 0.0458 |
| LR | 0.8837 | 0.4679 | 0.9639 | 1.0421 |
| SVM (10,000 samples) | 0.877 | 0.4261 | 0.9781 | 155.509 |

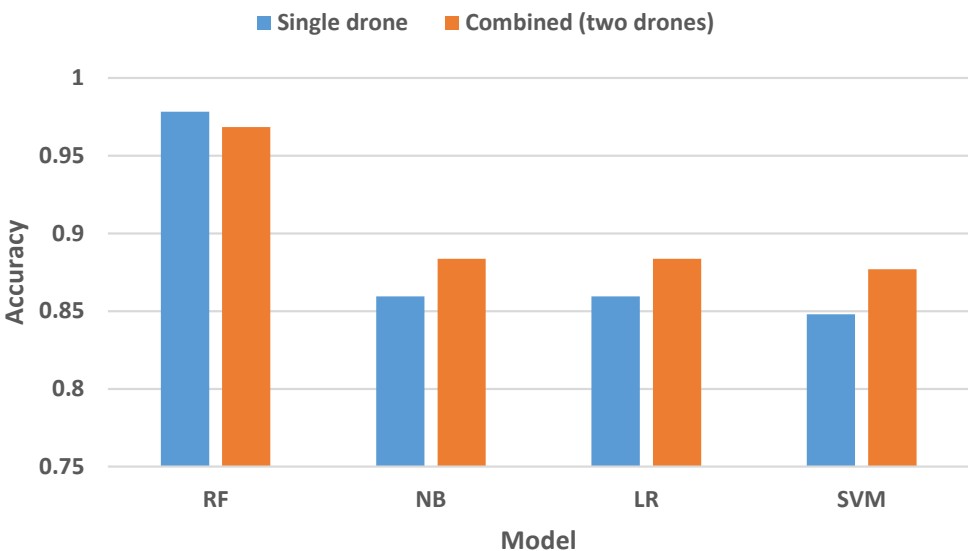

**Figure 13.** Comparison of model accuracy for datasets obtained from (blue) single and (orange) combined (two drones).

## 6. Conclusions

The performance evaluation of UAV attack detection was performed using various machine learning algorithms. The results showed that random forest had superior detection capabilities. In future work, we would like to include more attack types as well include flight logs from different drones. The current dataset contains flight logs only from single drone and hence the developed models must be trained on dataset from different drones to better generalize.

**Author Contributions:** Conceptualization, Z.B. and N.M.; methodology, N.S. and Z.B.; validation, N.M. and Z.B.; project administration, Z.B and N.M.; Experimentation and Analysis, N.S. All authors have read and agreed to the published version of the manuscript

**Funding:** This research was funded by Prince Mohammad Bin Fahd University Futuristic Center Grant.

**Institutional Review Board Statement:** Not Applicable.

**Informed Consent Statement:** Not Applicable.

**Data Availability Statement:** The researchers would like to acknowledge Deakin University and Prince Mohammad Bin Fahd University (PMU Futuristic Center Grant) for their continuing support for research and development.

**Conflicts of Interest:** The authors declare no conflict of interest.

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
