# Peer review of "Securing the Smart City Airspace: Drone Cyber Attack Detection through Machine Learning"

_futureinternet, doi:10.3390/fi14070205_

Round 1
Reviewer 1 Report
The topic of the study - the protection of drones from cyber threats is certainly relevant. The approach proposed by the authors based on machine learning is promising. The data set used is somewhat narrow (28 flight logs from a single drone) to draw a definitive conclusion on the effectiveness of the proposed method under real conditions. But it must be taken into account that this is an initial study that the authors plan to continue with a large set of data and drone models. Given this, the manuscript may be published.
The review of the literature (22 sources) is narrow given the topic raised in the study and could be expanded by the authors for a greater comparison of the results obtained with other works.
Author Response
Reviewer #1 Comments and Responses
The topic of the study - the protection of drones from cyber threats is certainly relevant. The approach proposed by the authors based on machine learning is promising.
Comments and suggestions
Comment #1: The data set used is somewhat narrow (28 flight logs from a single drone) to draw a definitive conclusion on the effectiveness of the proposed method under real conditions. But it must be taken into account that this is an initial study that the authors plan to continue with a large set of data and drone models. Given this, the manuscript may be published.
Response: We thank the respected reviewer for identifying this issue in regards to a narrow dataset comprising flight logs obtained from a single drone. In order to expand the substantiation of a machine learning based attack detection scheme, we have now incorporated the flight log data obtained from a 2nd drone of the same DJI Phantom4 family. The results of the machine learning based analysis when applied to the 2nd drone dataset have now been included in the paper to strengthen our findings.
Comment #2: The review of the literature (22 sources) is narrow given the topic raised in the study and could be expanded by the authors for a greater comparison of the results obtained with other works.
Response: Indeed the numbers of references that were included in this study were quite limited. This has now been expanded to include 29 references all together.
Reviewer 2 Report
The paper describes a machine-learning based detection system for different attacks that can be carried out against a drone. Such attacks include Denial of Service (DoS), deliberate motor shutdown and GPS signal jamming.
My main concern with this work is that on the one hand, all the data is collected from a single device, so it is quite unclear how general this approach is, and on the other hand, all the “attacks” have been introduced in the dataset based on the intuitions of the authors. For example, sudden drops in motor speed are consider “potentially anomalous”, or attacks where the drone might have been hijacked and the attacker is trying to make it crash. Note that such events are already present in the data collected from the drone, and are consequently part of its regular or expected behavior. The same comment applies for the rest of the considered anomalies.
In a nutshell, if normal data that has some anomalies is used, and the anomalies are labeled manually by the authors as attacks since they theoretically match what one could expect from such attacks, it is completely unclear if actual normal behavior will be incorrectly classified as attack or not.
One of the inputs for the detection system is packet loss, how would packet loss be measured in a real world scenario? This is relevant to me since the detection algorithms (or at least that is my understanding) will run in the drone, and it is unclear how could it be aware of the expected or actual packet loss rate.
Since all the data referring to packet lost was introduced by the authors, and it is used for detecting DoD, it is not fair to justify the detection problems showed by some of the algorithms by saying that losses could be due to other non-adversarial causes and therefore it is a tricky situation to detect.
Finally, the paper would really benefit from further proofreading.
Author Response
Reviewer #2 Comments and Responses
The paper describes a machine-learning based detection system for different attacks that can be carried out against a drone. Such attacks include Denial of Service (DoS), deliberate motor shutdown and GPS signal jamming.
Comment #1: My main concern with this work is that on the one hand, all the data is collected from a single device, so it is quite unclear how general this approach is,
Response: We thank the respected reviewer for identifying this issue in regards to the limitation of this study as it comprises flight logs obtained from a single drone. In order to expand the substantiation of a machine learning based attack detection scheme, we have now incorporated the flight log data obtained from a 2nd drone of the same DJI Phantom4 family. The results of the machine learning based analysis when applied to the 2nd drone dataset have now been included in the paper to strengthen our findings.
Comment #2: On the other hand, all the “attacks” have been introduced in the dataset based on the intuitions of the authors. For example, sudden drops in motor speed are consider “potentially anomalous”, or attacks where the drone might have been hijacked and the attacker is trying to make it crash. Note that such events are already present in the data collected from the drone, and are consequently part of its regular or expected behavior. The same comment applies for the rest of the considered anomalies.
Response: The writeup was probably not imparting the effect of specific drone flight parameters that refer to a cyber attack (GPS signal jamming and DoS attacks). We have now expanded upon these two attacks through inclusion of references that have discussed these attacks against drones in the past. Consequently, evidential backing of the types of attacks that we are aiming to model and present as part of the dataset is now valid. Indeed the drone datasets that were studies and analysed in this paper were
Comment #3: In a nutshell, if normal data that has some anomalies is used, and the anomalies are labeled manually by the authors as attacks since they theoretically match what one could expect from such attacks, it is completely unclear if actual normal behavior will be incorrectly classified as attack or not.
Response: Indeed normal behaviour refers to routine drone flights. The associated parametric values will be within a given value range. If these values change abruptly or go beyond the acceptable range of values, then these are referred to as anomalies. In the context of our proposed scheme, these parameters include: osd_data:connectedToRC’, ’RC_Info:frame_lost and rc_connect
Jamming of radio signals and spoofing of the same is a significant issue for drones, referring to a potential drop in the connection with the ground controller.
We have also identified the fact that DoS attacks are hard to differentiate from routine flash crowd traffic even for legacy systems. The outcome remaining the same (loss of system resources and sustenance of the flight).
We have now significantly enhanced the content of Section 3.2, to provide a succinct presentation of the 3 attacks being considered and how these affect the parametric values of various fields of a drone’s data log.
Comment #4: One of the inputs for the detection system is packet loss, how would packet loss be measured in a real world scenario? This is relevant to me since the detection algorithms (or at least that is my understanding) will run in the drone, and it is unclear how could it be aware of the expected or actual packet loss rate.
Response: Packet loss rates were not in the scope of the proposed scheme. We have now excluded the sentence ‘A possible DoS attack against a drone can cause the data communication flow between 349 a ground controller and the drone to notice significant packet loss,’ from the manuscript.
The whole paragraph on page 10 (last 2 paragraphs of Section 3) have now been rewritten to better explain the attack data and how this has been construed through: 1) labelling of the existing dataset and 2) through introduction of new data samples referring to attacks into the original dataset
Comment #5: Since all the data referring to packet lost was introduced by the authors, and it is used for detecting DoD, it is not fair to justify the detection problems showed by some of the algorithms by saying that losses could be due to other non-adversarial causes and therefore it is a tricky situation to detect.
Response: Please refer to the response for Comment #4, we have now excluded the misleading term of ‘packet loss’ from the draft. However, the proposed machine learning based classification is able to detect both malicious attempts to disrupt drone flights through attacks such as GPS signal jamming and DoS, as well as naturally occurring phenomena that may cause abrupt disruption of flight (including lightning strikes, strong winds and heavy rain).
Comment #6: Finally, the paper would really benefit from further proofreading.
Response: We have now done a thorough proofreading of the manuscript and the resulting draft is a significantly better read as compared to the previous version.